# NF-kB Regulation and the Chaperone System Mediate Restorative Effects of the Probiotic *Lactobacillus fermentum* LF31 in the Small Intestine and Cerebellum of Mice with Ethanol-Induced Damage

**DOI:** 10.3390/biology12111394

**Published:** 2023-11-01

**Authors:** Letizia Paladino, Francesca Rappa, Rosario Barone, Filippo Macaluso, Francesco Paolo Zummo, Sabrina David, Marta Anna Szychlinska, Fabio Bucchieri, Everly Conway de Macario, Alberto J. L. Macario, Francesco Cappello, Antonella Marino Gammazza

**Affiliations:** 1Department of Biomedicine, Neurosciences and Advanced Diagnostics (BiND), University of Palermo, 90127 Palermo, Italy; francesca.rappa@unipa.it (F.R.); rosario.barone@unipa.it (R.B.); francescopaolozummo@gmail.com (F.P.Z.); fabio.bucchieri@unipa.it (F.B.); francesco.cappello@unipa.it (F.C.); 2Euro-Mediterranean Institute of Science and Technology (IEMEST), 90139 Palermo, Italy; econwaydemacario@som.umaryland.edu (E.C.d.M.); ajlmacario@som.umaryland.edu (A.J.L.M.); 3Institute of Translational Pharmacology (IFT), Italy National Research Council of Italy (CNR), 90146 Palermo, Italy; 4Department of SMART Engineering Solutions & Technologies, eCampus University, 22060 Novedrate, Italy; fil.macaluso@gmail.com; 5Department Surgical, Oncological and Oral Sciences, School of Medicine, University of Palermo, 90133 Palermo, Italy; sabrina.david@unipa.it; 6Faculty of Medicine and Surgery, UKE-Kore University of Enna, Cittadella Universitaria, 94100 Enna, Italy; martaanna.szychlinska@unikore.it; 7Department of Microbiology and Immunology, School of Medicine, University of Maryland, Baltimore-Institute of Marine and Environmental Technology (IMET), Baltimore, MD 21202, USA

**Keywords:** gut microbiota, gut-brain axis, *Lactobacillus fermentum*, ethanol, chaperone system, heat shock proteins

## Abstract

**Simple Summary:**

Probiotics are live microorganisms that offer health benefits, primarily by improving the gut microbiota. A study involving mice explored the impact of *Lactobacillus fermentum* LF31 (*L. fermentum*), a probiotic, when combined with ethanol consumption over 12 weeks. The research found that *L. fermentum* influenced the NF-κB signaling pathway in the small intestine, inducing the expression of Hsp60, Hsp90, and IkB-α. It also suppressed the expression of pro-inflammatory molecules like IL-6 and TNF-α. These effects coincided with the restoration of the intestinal barrier damaged by ethanol, including the production of tight junction proteins, reducing the ethanol-induced intestinal permeability. Notably, the benefits of *L. fermentum* extended to the cerebellum, where it downregulated markers associated with glial inflammation. This indicates that the probiotic has anti-inflammatory and cytoprotective properties, not only in the gut but also in the brain, mitigating ethanol-induced gut permeability and neuroinflammation. Overall, this study suggests that *L. fermentum* may be a valuable tool in treating intestinal diseases caused by factors such as inflammation and dysbiosis and could be considered for therapeutic use in combination with other therapies for such conditions.

**Abstract:**

Probiotics are live microorganisms that yield health benefits when consumed, generally by improving or restoring the intestinal flora (microbiota) as part of the muco-microbiotic layer of the bowel. In this work, mice were fed with ethanol alone or in combination with the probiotic *Lactobacillus fermentum* (*L. fermentum*) for 12 weeks. The modulation of the NF-κB signaling pathway with the induction of Hsp60, Hsp90, and IkB-α by the probiotic occurred in the jejunum. *L. fermentum* inhibited IL-6 expression and downregulated TNF-α transcription. NF-κB inactivation concurred with the restoration of the intestinal barrier, which had been damaged by ethanol, via the production of tight junction proteins, ameliorating the ethanol-induced intestinal permeability. The beneficial effect of the probiotic on the intestine was repeated for the cerebellum, in which downregulation of glial inflammation-related markers was observed in the probiotic-fed mice. The data show that *L. fermentum* exerted anti-inflammatory and cytoprotective effects in both the small intestine and the cerebellum, by suppressing ethanol-induced increased intestinal permeability and curbing neuroinflammation. The results also suggest that *L. fermentum* could be advantageous, along with the other available means, for treating intestinal diseases caused by stressors associated with inflammation and dysbiosis.

## 1. Introduction

Intestinal permeability refers to the easy passage of substances through the intestinal wall. When the tight junctions (TJs) of the intestinal wall loosen (“leaky gut”), the gut becomes more permeable, allowing bacteria and toxins to pass from the gut into the bloodstream. Intestinal permeability is a recognized characteristic of several inflammatory and autoimmune diseases affecting the digestive system, including inflammatory bowel and celiac diseases [1]. There is emerging evidence of the link between gut microbiota and the central nervous system (CNS) physiology and pathology, underlined by the demonstrated influence of the gut microbiota on the blood–brain barrier function, which is a critical node within the microbiota–gut–brain axis [2,3].

Heat shock proteins (Hsps) are components of the Chaperon System that mediate the cellular stress response and represent an important class of evolutionarily conserved proteins in all three domains of life. These proteins are essential for maintaining protein homeostasis and promoting cell survival by preventing and counteracting the deleterious effects of stressors, for instance heat shock, oxidative stress, toxins, and diseases (e.g., bowel, breast, and thyroid cancers) [4,5,6].

In the gastrointestinal tract, Hsps have a pivotal role in preserving the integrity of the intestinal epithelial cell (IEC) barrier. The IEC barrier is essential for maintaining the separation between the gut lumen, which contains a complex microbiota and various antigens, and the host’s internal tissues. Alterations in this barrier function can lead to disorders such as dysregulated nutrient transport and increased permeability, which are associated with gut inflammation and various gastrointestinal pathologies. Hsps are involved in protecting IECs from stressors that can disrupt the intestinal barrier. These stressors may include factors like inflammation, infection, and changes in the gut microbiota. By assisting in the proper folding and stability of proteins within IECs, Hsps help to maintain the barrier function. Probiotics may have the ability to promote the expression of Hsps, which in turn exert their anti-inflammatory roles by inhibiting the NF-κB signaling pathway [7,8]. This finding suggests a probable modulatory crosstalk between the Hsps and NF-κB signaling pathways.

Probiotics could play a key role against intestinal inflammation by modulating the gut microbiota composition and producing protective bioactive metabolites [9]. For instance, lactic acid-producing bacteria are the most commonly used probiotics that play an important role in protecting the host against harmful microorganisms, strengthening the host immune system, improving feed digestibility, and reducing metabolic disorders, with significant improvements in the body mass index, blood pressure, glucose metabolism, and lipid profile [10,11,12]. Here, our attention was focused on *Lactobacillus fermentum* (*L. fermentum*), a Gram-positive bacterium belonging to the genus *Lactobacillus*, which can enhance the immunological responses as well as prevent community-acquired gastrointestinal and upper respiratory infections [10,13].

In this study, *L. fermentum* LF31 (Bromatech S.r.l., Milan, Italy), a widely marketed dietary supplement that is part of a mixture of milk enzymes that promote gut flora balance, was introduced in the diet of mice subjected to chronic ethanol (EtOH) exposure. The activation of the NF-κB transduction pathway in explanted jejunum and cerebellum samples was investigated to determine the mechanisms involved in the putative protective effects of *L. fermentum* against oxidative stress and inflammation.

## 2. Materials and Methods

### 2.1. Animal Model

Female 12-month-old mice (BALB/cAnNHsd) obtained from Harlan laboratories S.r.l. (Udine, Italy) were treated as previously described (Figure 1) [14].

All animal procedures and care were approved by the Italian Ministry of Health (authorization number: 1190/2020-PR) and performed in accordance with national and EU guidelines for the handling and use of experimental animals. Briefly, the animals were housed at the ATeN Center laboratories, University of Palermo, Italy, and kept at a constant 12:12 h light–dark cycle and had free access to food (4RF21 standard diet, Mucedola; Settimo Milanese, Milan, Italy) and water, which was the diet for the control (Ctrl) mice (*n* = 5). For the experimental mice, ethanol 96% (EtOH) (Girolamo Luxardo, Torrella, Padova, Italy) was added in the diet to account for 15% of total calories [15,16]. A calculation to increase the standard caloric need of the mice (14 kcal per day, 3.6 g of their food) with the addition of EtOH was made considering the caloric content of EtOH, diluting it in water in a final volume of 100 μL. This volume was orally administered to the mice once a day using a pipette for 12 weeks (12 EtOH, *n* = 5).

Another group of mice (12 EtOH + Probiotic, *n* = 5) were given EtOH as the others, but they also received orally the probiotic *Lactobacillus fermentum* LF31 (Bromatech S.r.l., Milan, Italy) 10^9^ CFUs (colony-forming units) per animal in water, each day during the week, with two days off on weekends, until the end of the experiment. The probiotic was given 20 min after the EtOH administration to investigate its efficacy to counteract the deleterious effects of ethanol [14]. The mice were euthanized 48 h after the last treatment and their brains and intestines were then dissected, fixed in formalin, and embedded in paraffin for analyses.

### 2.2. Histopatology

Cerebellum and jejunum tissue sections with a thickness of 5 μm were obtained from paraffin blocks and stained with hematoxylin and eosin (H&E) for histological examinations, such as length and width of the intestinal villi, the depth of the glandular crypts, and the thickness of the entire intestinal wall as well as the distribution of goblet cells. Sections were de-waxed in xylene for 10 min and rehydrated by sequential immersion in decreasing ethanol concentrations. The sections were then stained with H&E and examined using an optical microscope (Microscope Axioscope 5/7 KMAT, Carl Zeiss, Milan, Italy) connected to a digital camera (Microscopy Camera Axiocam 208 color, Carl Zeiss, Milan, Italy) for morphological analysis. Cerebellum and jejunum tissue sections were evaluated by two independent observers (F.R. and F.C.) in a blind manner using coded slides.

### 2.3. Immunofluorescence

Immunofluorescence was performed as previously described. Briefly, cerebellum and jejunum sections were incubated in the antigen unmasking solution (10 mM tri-sodium citrate, 0.05% Tween-20) for 8 min at 75 °C and treated with a blocking solution (3% BSA in PBS) for 30 min. Afterwards, the primary antibody (for specifications, see Table 1) was applied and the sections were incubated in a humidified chamber at 4 °C overnight. The following day, the sections were incubated for 1 h at room temperature (24 °C) with a conjugated species-specific fluorescent secondary antibody (dilution 1:100, anti-mouse ATTO-488and anti-rabbit ATTO-647, Sigma-Aldrich, St. Louis, MO, USA; anti-rabbit IgG–FITC antibody produced in goat, F0382), while nuclei were stained with DAPI (1:1000, Sigma-Aldrich) for 8 min. Images were captured using a Leica Confocal Microscope TCS SP8 (Leica Microsystems, Wetzlar, Germany).

### 2.4. Immunohistochemistry

Immunohistochemistry was performed as previously described [17] on 5 μm thick sections of paraffin-embedded jejunum. Sections were immunostained using the Histostain^®^-Plus 3rd Gen IHC Detection Kit, which utilized the labeled biotin-streptavidin methodology. The primary antibodies used are reported in Table 1. Appropriate positive and negative (isotype) controls were run concurrently. Nuclear counterstaining was carried out using hematoxylin (Hematoxylin aqueous formula, N. Cat. S2020, DAKO). Finally, the slides were prepared for observation with coverslips using a permanent mounting medium (Vecta Mount, H-5000, Vector Laboratories, Inc., Burlingame, CA, USA). Observation of the sections was performed with an optical microscope (Microscope Axioscope 5/7 KMAT, Carl Zeiss, Milan, Italy) connected to a digital camera (Microscopy Camera Axiocam 208 color, Carl Zeiss, Milan, Italy). Two independent observers (F.C. and F.R.) evaluated the reactions on two separate occasions and performed a semi-quantitative analysis to determine the percentage of immunopositivity. All observations were conducted at a magnification of 400× and the percentage of positive cells was calculated in a high-power field (HPF) and repeated for 10 HPFs. The arithmetic mean of counts was used for statistical analysis.

### 2.5. RNA Preparation and Quantitative Real Time Polymerase Chain Reaction (RT-PCR) Assay

Total RNA was extracted using the Tri-Reagent (Sigma Aldrich, Milan, Italy) according to the manufacturer’s instructions, from 500 mg of cryopreserved intestinal tissues of control and treated mice. Approximately 300 ng of RNA, determined spectrophotometrically, was retrotranscribed using the ImProm-II Reverse Transcriptase System (Promega Corporation, Madison, WI, USA) to obtain cDNA, which was amplified using the STepOnePlus Real-Time PCR System (Life Technologies, Carlsbad, CA, USA). qRT-PCR analysis was performed using GoTaq qPCR Master Mix (A6001, Promega Corporation). All samples were run in triplicate. A threshold cycle (CT) was observed in the exponential phase of amplification, and quantification of relative expression levels was performed with standard curves for target genes and the endogenous control (GAPDH). Geometric means were used to calculate the ΔΔCT (delta-delta CT) values and expressed as 2^−ΔΔCT^. Complementary deoxyribonucleic acid (cDNA) was amplified using the Rotor-gene 6000 Real-Time PCR Machine (Qiagen GmbH, Hilden, Germany), with the primers indicated in Table 2.

### 2.6. Statistical Analysis

One-way ANOVA followed by the Bonferroni post hoc test for multiple comparisons were applied for data analysis. All statistical analyses were performed using the GraphPad Prism^TM^ 4.0 program (GraphPad Software Inc., San Diego, CA, USA). All data are presented as mean ± SD, and the level of statistical significance was set at *p* < 0.05.

## 3. Results

### 3.1. Small Intestinal Histomorphometry

The morphological analysis conducted on the mucosa of the jejunum slices stained with H&E was conducted to evaluate the length and width of the intestinal villi, the depth of the glandular crypts, and the thickness of the entire intestinal wall. We also observed the goblet cells present in the villous lining epithelium (Figure 2 and Appendix A).

The data obtained show a statistically significant difference in the length of the villi. The average length of the villi was up to 380 µm, 600 µm, and 500 µm, respectively for Ctrl (Figure 2A), 12 EtOH (Figure 2B), and 12 EtOH + Probiotic (Figure 2C) mice. The length of villi significantly increased in 12 EtOH and 12 EtOH + Probiotic compared with the Ctrl, while it decreased in 12 EtOH + Probiotic compared to 12 EtOH (Figure 2D, *p* < 0.001). No statistically significant differences were observed in the width of the villi and the depth of the glandular crypts (Appendix A). The intestinal wall thickness, probably as a consequence of the villus length difference, was significantly greater in the 12 EtOH and 12 EtOH + Probiotic groups than in the control group (*p* < 0.001) and between the 12 EtOH and 12 EtOH + Probiotic groups (Appendix A, *p* < 0.05). A reduction in the number of goblet cells, one of the major components of the intestinal barrier producing mucin, was also observed in the 12 EtOH (Figure 2F) and 12 EtOH + Probiotic (Figure 2G) groups compared with the Ctrl (Figure 2E; black arrows indicate goblet cells).

### 3.2. Effects of L. fermentum Administration on Hsp60 and Hsp90 Immunoreactivity in the Jejunum

An immunomorphological evaluation, using immunohistochemistry, was performed to determine Hsp60 and Hsp90 tissue levels and immunolocalization, since these proteins are classically involved in the stress response. As described later, Hsp60 and Hsp90 immunoreactivity decreased in the enterocytes lining the intestinal villi in the jejunum from the 12 EtOH group (Figure 3B,F) compared to the Ctrl (Figure 3A,E) and 12 EtOH + Probiotic (Figure 3C,G) groups. Immunopositivity for Hsp60 was diffuse cytoplasmic and sometimes granular, while Hsp90 immunolocalization was widespread cytoplasmic. The semi-quantitative analysis confirmed that the decrease was statistically significant (Figure 3D,H, respectively, *p* < 0.001).

### 3.3. Effects of Ethanol Consumption and L. fermentum Administration on the Intestinal Barrier Integrity and Permeability

To determine the effects of ethanol and *L. fermentum* administration on the intestinal barrier integrity, the tissue levels and immunolocalization of the matrix zonula occludens-1 (ZO-1), IL-6, and metalloproteinase 9 (MMP-9) were determined using immunohistochemistry and immunofluorescence, while the expression of TNF-α was investigated using qRT-PCR.

Immunofluorescence was used to determine ZO-1 levels in the intestinal barrier. The immunoreactivity for this junctional protein decreased in the jejunum samples of the 12 EtOH (Figure 4B) group compared with the Ctrl (Figure 4A) and 12 EtOH + Probiotic (Figure 4C) groups. In the samples from Ctrl and 12 EtOH + Probiotic mice, ZO-1 immunopositivity was uniformly localized in the cytoplasm and in the membrane of the epithelial cells lining the intestinal villi, while the 12 EtOH samples showed altered distribution patterns for the protein. The same techniques were used to determine IL-6 immunoreactivity, that were lower in the Ctrl (Figure 4D) and 12 EtOH + Probiotic groups (Figure 4F) compared to the 12 EtOH group (Figure 4E).

As shown in Figure 5, immunopositivity for MMP-9 was higher in intensity in the cytoplasm of the enterocytes in the 12 EtOH (Figure 5B) group compared to the Ctrl (Figure 5A) and 12 EtOH + Probiotic groups (Figure 5C) and in the 12 EtOH + Probiotic group compared to the Ctrl. The average percentage of immunopositivity was 26,6%, 78,6%, and 53,4% for the Ctrl, 12 EtOH, and 12 EtOH + Probiotic groups, respectively, and these differences were statistically significant (Figure 5D, *p* < 0.001).

The TNF-α gene expression level significantly increased in the 12 EtOH group compared to the control (*p* < 0.03), while the difference between the 12 EtOH and 12 EtOH + Probiotic groups was statistically non-significant (Figure 5E).

### 3.4. Effect of L. fermentum on pNF-kB and IkB-α Tissue Distribution

To determine the molecules involved in the effects of *L. fermentum* on intestinal barrier integrity and permeability, the tissue levels of phospho-NFkB p65 (pNF-kB) and IkB-α (NF-κB inhibitor) were investigated using immunohistochemistry and immunofluorescence, respectively, while the expression of TNF-α was investigated using qRT-PCR.

The immunohistochemical stain of pNF-kB revealed a stronger nuclear signal in the 12 EtOH group (Figure 6B) compared to the Ctrl (Figure 6A) and 12 EtOH + Probiotic (Figure 6C) groups. The semi-quantitative analysis revealed a significant increase in pNF-kB nuclear positivity in the 12 EtOH group compared to the 12 EtOH + Probiotic and Ctrl groups (*p* < 0.001 and *p* < 0.05, respectively). The probiotic induced a significant decrease in pNF-κB nuclear distribution compared to 12 EtOH (Figure 6D, *p* < 0.001).

Concurrently, IkB-α distribution was low in the 12 EtOH group (Figure 7B) compared to the Ctrl (Figure 7A) and 12 EtOH + Probiotic groups (Figure 7C).

### 3.5. Effects of Ethanol Consumption and L. fermentum Administration on the Cerebellum: Morphological Observations and Determination of Astrocyte Activation

H&E staining was conducted on cerebellum slices to evaluate whether the EtOH intake caused morphological changes (Figure 8). The histological sections of the cerebellum of the 12 EtOH (Figure 8B) group mice showed the presence of vacuolization in white matter (WM) that was less present after treatment with the probiotic (red arrows). A change in the morphology of Purkinje cells (PCs) was also observed; many of these cells showed an altered morphology, often round rather than pear-shaped, and their nucleus lacked a prominent nucleolus. The latter features were observed in the 12 EtOH (Figure 8E) and 12 EtOH + Probiotic (Figure 8F) groups (black arrows).

GFAP and S100B were used as markers of neuronal inflammation and astrogliosis. The immunofluorescence analysis revealed that the mice presented GFAP immunoreactive astrocytes in the WM. The astrocytes of EtOH-treated (Figure 9B) mice showed increased numbers of positive GFAP, while GFAP immunoreactivity decreased in the 12 EtOH + Probiotic group (Figure 9C). S100B immunoreactivity was strong in the 12EtOH (Figure 9E) and 12 EtOH + Probiotic groups (Figure 9F), compared to the Ctrl (Figure 9D), and increased in the soma of Bergmann glial cells within the Purkinje cell layer of EtOH-treated mice in comparison with the Ctrl mice (white arrows).

### 3.6. In L. fermentum-fed Mice the Cerebellum Showed Increased Hsp60 and Hsp90 and Decreased pNF-κB Immunoreactivity

Hsp60, Hsp90, and pNF-κB immunoreactivity were also evaluated on the cerebellum slices of the mice using immunofluorescence. As in the jejunum, both Hsp60 and Hsp90 levels increased in the 12 EtOH + Probiotic (Figure 10C,F, respectively) group compared to the 12 EtOH group (Figure 10B,E, respectively). The two proteins appeared in the neuropil and in the body of cells with a neuron-like morphology.

The immunoreactivity for pNF-κB was low in the Ctrl (Figure 11A) group and the 12 EtOH + Probiotic group (Figure 11C), while it increased in the 12 EtOH group (Figure 11B), in particular in the WM and granular layer area (not shown).

## 4. Discussion

The muco-microbiotic layer of the bowel hosts the microbiota and all the soluble factors that permit a crosstalk between the corpuscular elements of the microbiota and the epithelial cells of the mucosa. In this study, the effects of *L. fermentum* against oxidative stress damage and inflammation induced by chronic alcohol intake was investigated in the jejunum and the cerebellum. The results obtained show that alcohol consumption induced a reduction of goblet cells and elongation of the villi, probably altering intestinal permeability while the probiotic preserved the intestinal barrier structure and villus morphology. Moreover, alcohol causes intestinal damage and triggers the over-expression of molecules involved in the stress response. The beneficial consequence of intestinal barrier preservation was also reflected by the reduction in neuroinflammatory markers. To the best of our knowledge, this is the first work exploring the expression of Hsp60 and Hsp90 in the small intestine and cerebellum in relation to probiotic intake in a mouse model of ethanol abuse. Ethanol was used as an oxidative stress inducer and Hsp60 and Hsp90 over-expression in the small intestine was paralleled in the cerebellum, as protective agents against oxidative stress, outlining a gut–brain axis.

Our attention was focused on the NF-κB pathway that was analyzed with the purpose of identifying a molecular mechanism underpinning the protective effects of the probiotic. Dysbiosis promotes the translocation of intestinal bacterial components, such as lipopolysaccharide (LPS) and peptidoglycan (PGN), from the intestinal lumen to the systemic circulation and other organs, including the CNS. This event is probably associated with an increased functionality of the transcription factor NF-κB, which serves as a pivotal mediator of inflammatory responses [18,19]. Normally, an inactive NF-κB heterodimer is bound to IκB in the cytosol. Activation of the inflammatory cascade causes phosphorylation and degradation of IκB, releasing NF-κB [19], which translocates to the nucleus and induces the expression of several factors, including pro-inflammatory cytokines [20,21]. In the present study, EtOH induced an increase in the immunoreactivity of IL-6 (Figure 4), MMP-9 (Figure 5), and pNF-κB (Figure 6) in the jejunum, while *L. fermentum* administration prevented their changes. Moreover, the probiotic promoted a down-regulation of the TNF-α gene expression level (Figure 5E), and these events were accompanied by an increased immunoreactivity of ZO-1 (Figure 4). MMP-9, IL-6, and TNF-α are pathogenic factors playing a crucial role in the progression of inflammatory responses in the gastrointestinal tract (for instance in patients with inflammatory bowel disease), causing an increase in intestinal permeability [22,23]. Therefore, our data suggest that the probiotic had a protective anti-inflammatory role, contributing to the preservation of the gut barrier integrity.

Alcohol-induced damage of the gut barrier may allow pathogen-associated molecules to escape from the intestinal lumen into the systemic circulation and other organs, including the CNS [24,25,26]. The effects of alcohol (e.g., inflammation and oxidative stress) may be reflected in the activation of several neurological pathways, causing changes in behavior and cognitive and memory deficits, amplifying neurodegeneration [9,27]. For this reason, the cerebellum samples were examined, and vacuolization was observed in the area corresponding to the WM in the cerebellum of EtOH-treated mice. This observation is in line with the reported vacuolization of cerebellar nuclei neurons and atrophy of the PCs typically associated with alcohol consumption and impaired neurotransmission [28]. Although neurons are the main conveyors of information between different brain regions, maintenance, survival, and normal activity of neurons are dependent on their interaction with glial cells [29]. As explained in detail in the results section, EtOH-treated mice showed GFAP-positive cells in the area presenting vacuoles, whereas GFAP immunoreactivity decreased in the cerebellum of the 12 EtOH + Probiotic group (Figure 9). Similar observations were made for S100B immunoreactivity, that was strong in the 12 EtOH and 12 EtOH + Probiotic groups compared to the Ctrl (Figure 9). The quantitative changes of these neuroinflammatory markers, together with the decrease in pNF-κB immunoreactivity, that we found in the cerebellum, support the notion that the probiotic also had anti-inflammatory activity in the CNS (Figure 11). To the best of our knowledge, this is the first in vivo study that suggests a correlation between *L. fermentum* intake and the modulation of GFAP and S100B in the cerebellum as markers of inflammation and tissue damage.

Hsps are induced in response to different types of stressors, such as oxidative stress, and are implicated in many pathways critical for maintaining tissue homeostasis [4,5,30,31]. In this work, Hsp60 and Hsp90 expression was investigated as a possible component of the mechanism involved in the protective effects of *L. fermentum* against alcohol insult in the intestine and, consequently, in the cerebellum. As detailed in the result section, Hsp60 and Hsp90 were strongly expressed in the jejunum (Figure 3) and cerebellum (Figure 10) of 12 EtOH + Probiotic mice, namely in those treated with the probiotic. It has been reported that in the gut, Hsps contribute to the maintenance of tight junctions between IECs and promote the barrier function [7,32,33]. We could not find reports of elevated expression of Hsp60 and Hsp90 in the cerebellum of mice fed with the probiotic, suggesting that no similar data are currently available in the literature, so the work described here appears to be the first to investigate the role of the two chaperones in the cerebellum in relation to ethanol-induced stress and the use of probiotics. It is tempting to speculate that an increased expression of these two proteins reflect their participation in mediating the anti-inflammatory and cytoprotective effects of *L. fermentum*, resulting in the inhibition of NF-κB activation in the cerebellum, similarly to what happened in the intestine. It has in fact been demonstrated that *L. fermentum* can inhibit NF-κB activation through the induction of some Hsps, such as Hsp25 and Hsp70, in colonic epithelial cells [33].

An important piece of evidence, supporting the existence of a molecular pathway involved in the effects of the probiotic, is provided by the immunofluorescence analysis of IkB-α (NF-κB inhibitor) in the jejunum. The results showed that IkB-α levels decreased in the 12 EtOH group and were restored in the 12 EtOH + Probiotic group, compared to the Ctrl group (Figure 7). Hsp60 and Hsp90 are probably able to form a complex with IkB kinase (IKK), inhibiting it and thus preventing NF-κB activation [34,35,36,37]. As mentioned above, the NF-κB pathway is a key signaling channel for the activation of immune responses secondary to a variety of stimuli and may represent a key link between probiotics, the gut, and the CNS, to build the brain–gut–microbiota axis [34]. Under non-stimulatory conditions, NF-κB is present in its inactive form in the cytoplasm, bound to the inhibitory molecule IκB [33,36,37]. When pro-inflammatory stimuli, such as alcohol consumption, trigger signaling pathways, IκB is phosphorylated and degraded by IKK. Once freed from IκB, NF-κB can migrate into the nucleus, bind target promoters, and activate the transcription of effector genes, influencing downstream cytokine secretion [38,39,40,41]. These events are in line with our finding regarding the regulation of IL-6 and TNF-α gene expression in the jejunum by the probiotic [42,43,44,45,46,47].

## 5. Conclusions

This study shows that the oral intake of *L. fermentum* LF31 can protect against alcohol-induced leaky gut injury in mice. The protective and anti-inflammatory effects exerted by the probiotic in the intestine may be reflected in the cerebellum as a consequence of the preservation of the intestinal barrier integrity. As a working hypothesis, it can be proposed that *L. fermentum* LF31induces Hsp60 and Hsp90 expression, thereby triggering inhibition of NF-κB activation, leading to IkB upregulation. Thus, preservation of the intestinal barrier integrity could occur via the modulation of MMP-9, IL-6, and TNF-α levels (Figure 12). A potential limitation of the study could be the small sample size since a larger sample size would have greater statistical power. However, the statistical methods used are designed to correct for false positives. Moreover, the use of smaller animal samples is consistent with ethical norms in biomedical research.

It appears that some of the molecules studied have potential to serve as biomarkers for monitoring the response to *L. fermentum* as a dietary supplement in the management of inflammatory disorders beyond those caused by ethilism, for instance, celiac disease.

## Figures and Tables

**Figure 1 biology-12-01394-f001:**
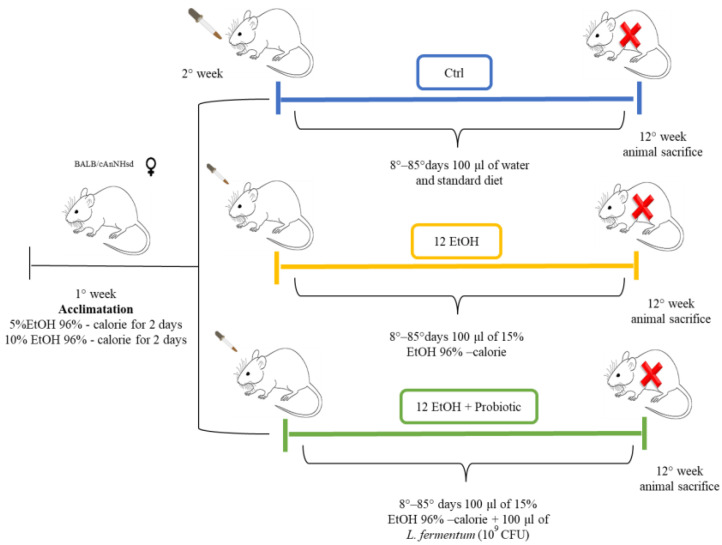
Animal model and diet. The control group (Ctrl, blue) received only the standard diet (without ethanol or probiotic) for the entire duration of the protocol (12 weeks). For the experimental mice, EtOH 96% was added to account for 15% of total calories in the diet. To acclimate the mice to the alcohol containing diet, they were given 5% EtOH-calorie content for 2 days, then 10% EtOH-calorie content for 2 days, and lastly 15% EtOH-calorie content for 12 weeks (12 EtOH, yellow). In addition, 12 EtOH + Probiotic (green) mice received orally the probiotic *L fermentum* LF31 (Bromatech S.r.l., Milan, Italy), 10^9^ CFU per animal per day, 5 days per week until the end of the experiments. The pertinent EtOH dose was diluted in water to a final volume of 100 μL. This volume was orally administered to the mice every day using a pipette and a tip, and after 20 min, the mice were fed with the probiotic in the same manner. At the end of the treatment the mice were euthanized (red X letter) and their brain and intestine were then dissected, fixed in formalin, and embedded in paraffin for further analyses [14]. Abbreviations: CFU, colony-forming units; Ctrl, control group; EtOH, ethanol; *L. fermentum* LF31, *Lactobacillus fermentum* LF31.

**Figure 2 biology-12-01394-f002:**
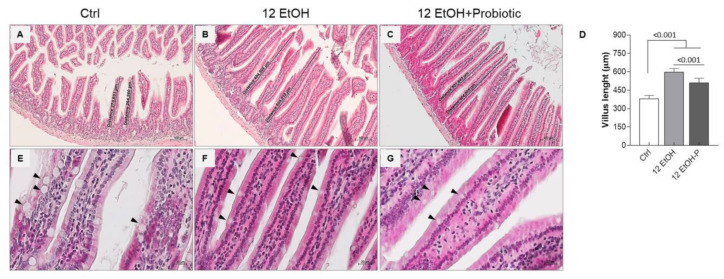
Morphological features of the mice jejunum. Representative images ((**A**), Ctrl; (**B**), 12 EtOH; (**C**), 12 EtOH + Probiotic) and histograms (**D**) of the H&E stain showing significant differences in the length of villi between the groups. Scale bar 100 µm. Data are presented as mean ± SD. Representative images of H&E stain displaying goblet cells (black arrows) distribution in the jejunum of Ctrl (**E**), 12 EtOH (**F**), and 12 EtOH + Probiotic (**G**) mice. Scale bar 20 µm. Ctrl: control, standard diet; 12 EtOH: 12 weeks of ethanol in the standard diet; 12 EtOH + Probiotic: 12 weeks of ethanol plus *L. fermentum* LF31 in the standard diet.

**Figure 3 biology-12-01394-f003:**
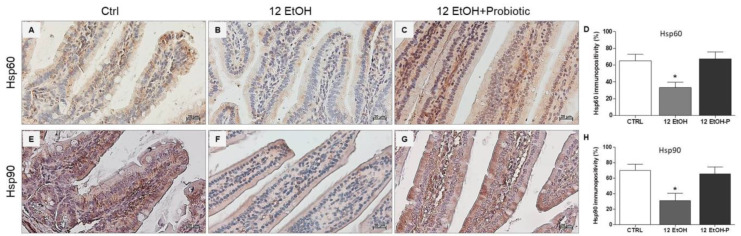
Representative images and histograms of Hsp60 (**A**–**D**) and Hsp90 distribution (**E**–**H**) in the jejunum of the mice. Magnification 400×, scale bar 20 µm. Data are presented as mean ± SD. Ctrl: control, standard diet; 12 EtOH: 12 weeks ethanol in the standard diet; 12 EtOH + Probiotic: 12 weeks ethanol plus *L. fermentum* LF31 in the standard diet. * Significantly different from Ctrl and 12 EtOH + Probiotic (*p* < 0.001).

**Figure 4 biology-12-01394-f004:**
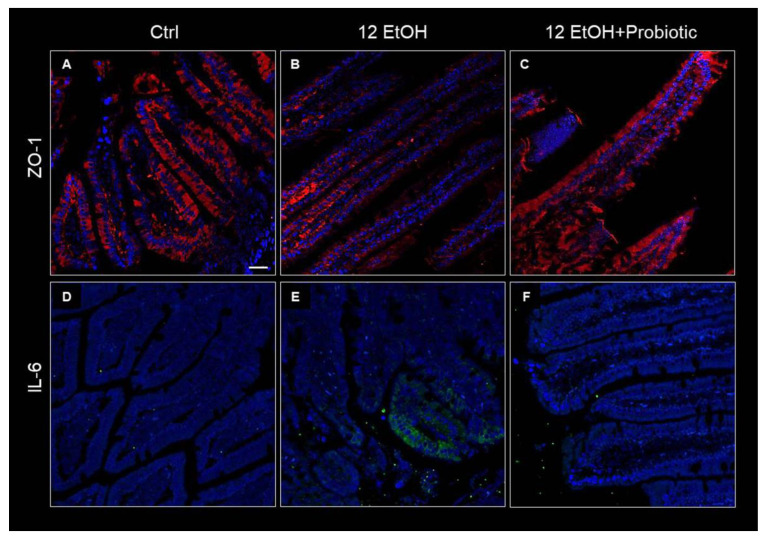
Immunofluorescence results for ZO-1 and IL-6. Representative images of ZO-1 (**A**–**C**) and IL-6 (**D**–**F**) distribution in the jejunum of Ctrl ((**A**,**D**), respectively), 12 EtOH ((**B**,**E**), respectively), and 12 EtOH + Probiotic (**C**,**F**), respectively) mice. Scale bar 25 µm. Ctrl: control, standard diet; 12 EtOH: 12 weeks of ethanol in the standard diet; 12 EtOH + Probiotic: 12 weeks of ethanol plus *L. fermentum* LF31 in the standard diet. Blue: nuclei; red: ZO-1; green: IL-6.

**Figure 5 biology-12-01394-f005:**
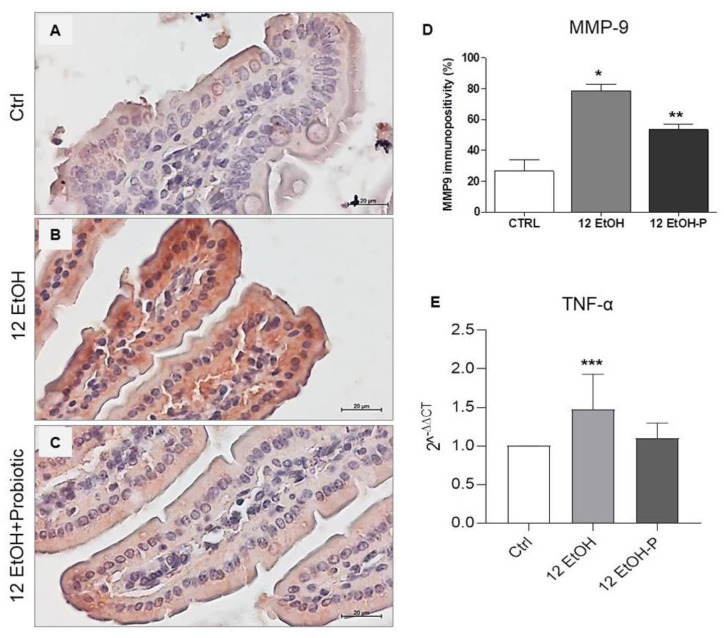
(**A**) Representative images of MMP−9 distribution in the jejunim of Ctrl (**A**), 12 EtOH (**B**), and 12 EtOH + Probiotic (**C**) mice. Magnification 400×, scale bar 20 µm. (**D**) Histograms showing the percentage of immunopositivity for MMP−9. Data are presented as mean ± SD. * Significantly different from Ctrl and 12 EtOH + Probiotic (*p* < 0.001); ** significantly different from Ctrl and 12 EtOH (*p* < 0.001). (**E**) Histograms showing TNF-α gene expression level. The bars indicate the degree of the cytokine gene expression level normalized for the reference genes, according to the Livak method (2^−ΔΔCT^). The ratio of pro-inflammatory cytokines levels/GAPDH levels was used as an indirect indication of TNF-α decrease. Data are presented as mean ± SD. *** Significantly different from Ctrl (*p* < 0.03). Ctrl: control, standard diet; 12 EtOH: 12 weeks of ethanol in the standard diet; 12 EtOH + Probiotic: 12 weeks of ethanol plus *L. fermentum* LF31 in the standard diet.

**Figure 6 biology-12-01394-f006:**
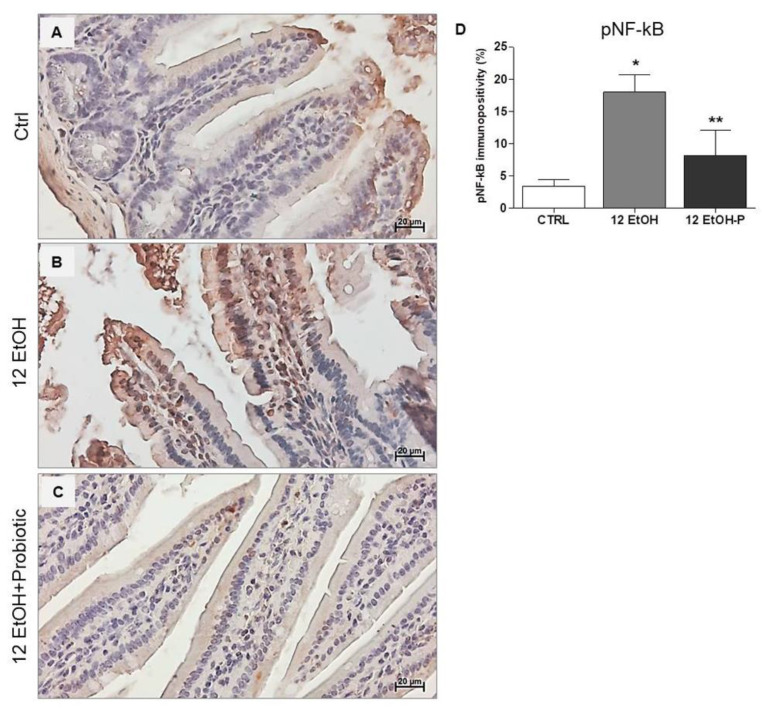
Immunohistochemical results for pNF-kB. Representative images of pNF-κB distribution in the jejunum of Ctrl (**A**), 12 EtOH (**B**), and 12 EtOH + Probiotic (**C**) mice. Magnification 400×, scale bar 20 µm. (**D**) Histograms showing the percentage of immunopositivity for pNF-κB. Data are presented as mean ± SD. Ctrl: control, standard diet; 12 EtOH: 12 weeks of ethanol in the standard diet; 12 EtOH + Probiotic: 12 weeks of ethanol plus *L. fermentum* LF31 in the standard diet. * Significantly different from Ctrl and 12 EtOH + Probiotic (*p* < 0.001); ** significantly different from Ctrl and 12 EtOH (*p* < 0.05).

**Figure 7 biology-12-01394-f007:**
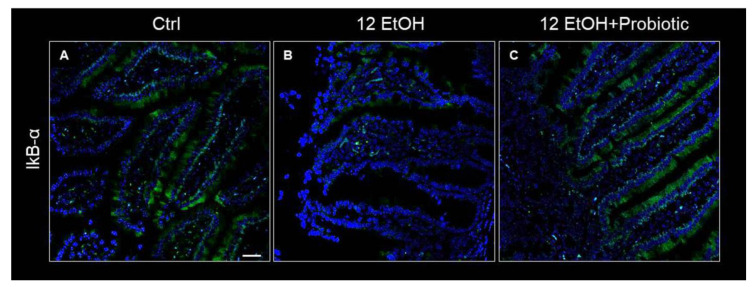
Representative IF images of IκB-α distribution on the jejunum of Ctrl (**A**), 12 EtOH (**B**), and 12 EtOH + Probiotic (**C**) mice. Scale bar 25 µm. Ctrl: control, standard diet; 12 EtOH: 12 weeks of ethanol in the standard diet; 12 EtOH + Probiotic: 12 weeks of ethanol plus *L. fermentum* LF31 in the standard diet. Blue: nuclei; green: IkB-α.

**Figure 8 biology-12-01394-f008:**
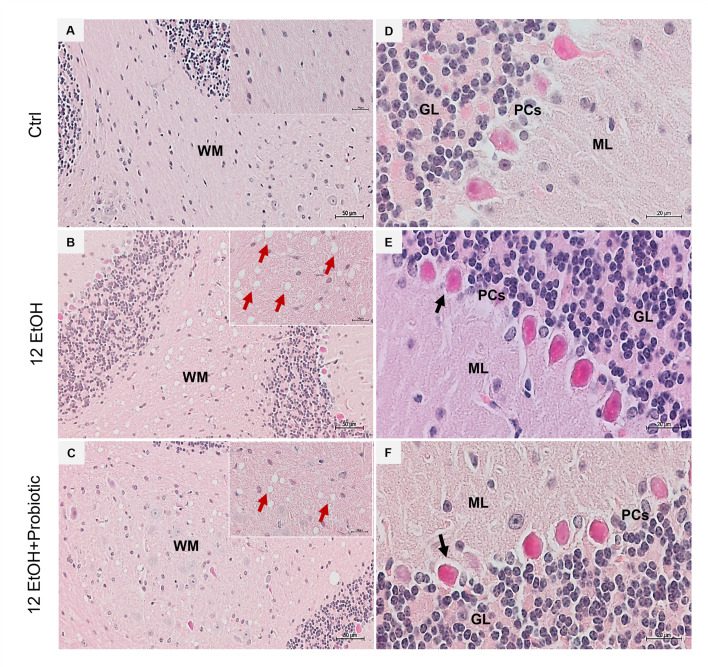
Histology of the cerebellum. Representative images of cerebellar H&E-stained sections from mice fed with a standard diet (Ctrl, (**A**,**D**)), ethanol (12 EtOH, (**B**,**E**)) diet, and ethanol plus *L. fermentum* LF31 diet (12 EtOH + Probiotic, (**C**,**F**)). On the left side column of images, vacuolization (red arrows) is shown in the area of subcellular white matter (WM) in the cerebellum of mice treated with ethanol alone or in combination with the probiotic. Magnification 20×, scale bar 50 µm. On the right-side column, black arrows indicate Purkinje cells with abnormal morphology. Magnification 630×, scale bar 20 µm. WM: white matter; ML: molecular layer; GL: granular layer; PCs: Purkinje cells.

**Figure 9 biology-12-01394-f009:**
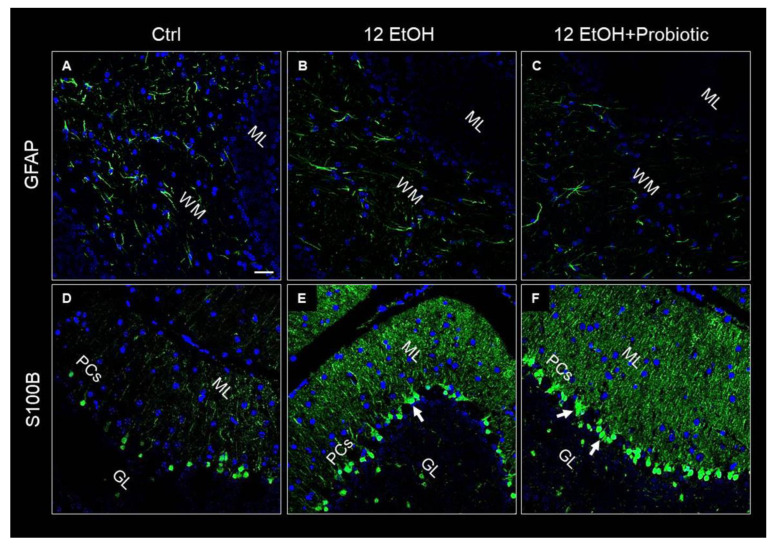
Representative IF images of GFAP (**A**–**C**) and S100B (**D**–**F**) tissue distribution on cerebellum slices of mice fed with a standard diet (Ctrl, (**A**,**D**), respectively), ethanol (12 EtOH, (**B**,**E**), respectively), and ethanol plus *L. fermentum* LF31 (12 EtOH + Probiotic, (**C**,**F**), respectively). Scale bar 25 µm. WM: white matter; ML: molecular layer; GL: granular layer; PCs: Purkinje cells. Blue: nuclei; green: GFAP and S100B.

**Figure 10 biology-12-01394-f010:**
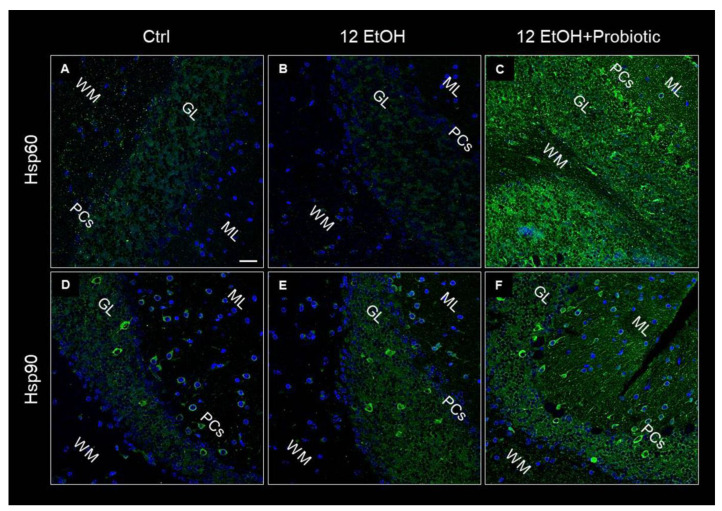
Representative images of Hsp60 (**A**–**C**) and Hsp90 (**D**–**F**) immunofluorescence results. The immunofluorescence reaction was conducted on cerebellum slices of mice fed with a standard diet (Ctrl, (**A**,**D**), respectively), ethanol (12 EtOH, (**B**,**E**), respectively), and ethanol *plus L. fermentum* LF31 (12 EtOH + Probiotic, (**C**,**F**), respectively). Scale bar 25 µm. WM: white matter; ML: molecular layer; GL: granular layer; PCs: Purkinje cells. Blue: nuclei; green: Hsp60 and Hsp90.

**Figure 11 biology-12-01394-f011:**
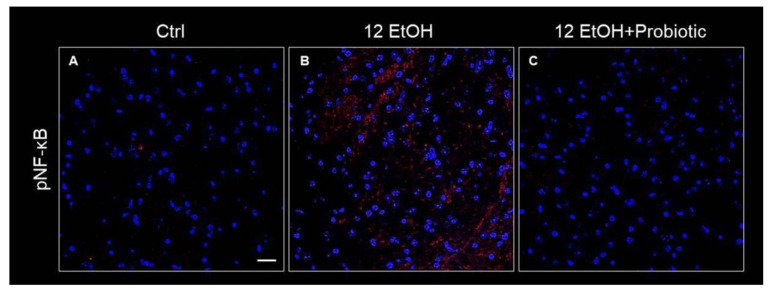
Immunofluorescence results for pNF-κB. Representative images of pNF-κB signal on the white matter of the cerebellum of mice fed with a standard diet (Ctrl, (**A**)), ethanol (12 EtOH, (**B**)), and ethanol and *L. fermentum* diets (12 EtOH + Probiotic, (**C**)). Scale bar 25 µm. Blue: nuclei; red: pNF-κB.

**Figure 12 biology-12-01394-f012:**
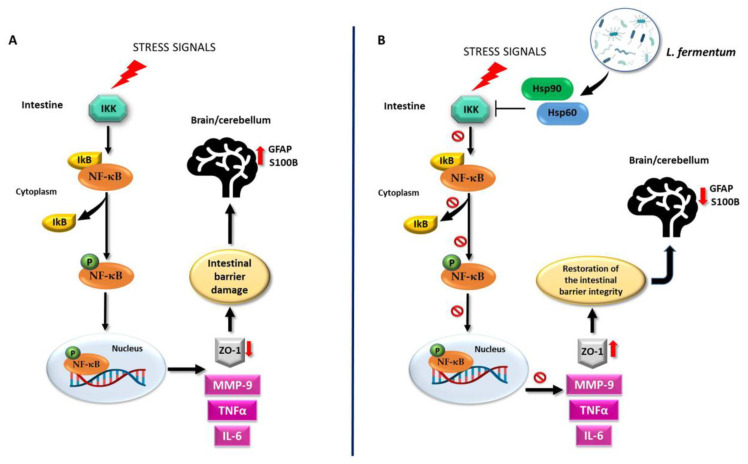
Working hypothesis. (**A**) Normally, the inactive NF-κB heterodimer is bound to its beta inhibitor (IκB) in the cytosol. Under stress signals, the degradation of IκB by IκB-kinase (IKK) induces the release and phosphorylation of NF-κB (pNF-κB). Activated pNF-κB moves to the nucleus and induces the expression of inflammatory cytokines, e.g., IL-6 and TNF-α, as well as that of the protease MMP-9. These events trigger the dysregulation of the intestinal barrier integrity via the impairment of ZO-1 expression. Inflammatory agents may then reach the systemic circulation and the central nervous system, for example, the cerebellum. There, because of the intestinal damage, GFAP and S100B become over-expressed, indicating neuroinflammation. (**B**) The molecular mechanisms associated with the putative properties of *L. fermentum* could be related to the NF-κB pathway. The probiotic may be able to induce an increase in Hsp60 and Hsp90 levels as a protective mechanism against oxidative stress. Hsp60 and Hsp90 could form a complex with IKK, inhibiting it and preventing the activation of NF-κB. In this manner, the integrity of the intestinal barrier can be restored through MMP-9, IL-6, and TNF-α down-regulation and ZO-1 over-expression. Consequently, the negative effects of alcohol-induced damage, such as the increase in neuroinflammatory markers in the cerebellum, can be ameliorated. It is possible that the regulation of the NF-κB pathway in the cerebellum following *L. fermentum* intake is like the one induced in the intestine, given the parallel modulation trend of the same molecules in both tissues.

**Table 1 biology-12-01394-t001:** Primary antibodies used in this study.

Primary Antibody	Source and Type	Supplier	Dilution	
			IHC ^1^	IF
Glial Fibrillary Acid Protein (GFAP)	Mouse monoclonal	Sc-33673—Santa Cruz, Biotechnology, Dallas, TX, USA	-	1:50
S100B-calcium-binding-protein (S100B)	Rabbit polyclonal	Z0311–Dako, Santa Clara, CA, USA	-	1:250
Phospho-proinflammatory nuclear factor (pNFκB)	Rabbit polyclonal	Sc-33020—Santa Cruz, Biotechnology, Dallas, TX, USA	1:100	1:50
Heat Shock Protein 60 (Hsp60)	Mouse monoclonal	Ab-13532—Abcam, Cambridge, UK	1:300	1:50
Heat Shock Protein 90 α/β (Hsp90 α/β)	Mouse monoclonal	Sc-13119—Santa Cruz, Biotechnology, Dallas, TX, USA.	1:200	1:50
Matrix Metalloproteinase 9 (MMP-9)	Rabbit polyclonal	Sc-10737—Santa Cruz Biotechnology, Dallas, TX, USA	1:100	-
Tight Junction Protein 1 (TJP1)	Rabbit polyclonal	C82740—Sigma Aldrich, Hamburg, Germany	-	1:50
NF-kappa-B inhibitor alpha (IκBα)	Rabbit polyclonal	Sc-203—Santa Cruz Biotechnology, Dallas, TX, USA	-	1:50
Interleukin 6 (IL-6)	Rabbit polyclonal	P620—Invitrogen, Carlsbad, CA, USA	-	1:50

^1^ Abbreviations: IHC, immunohistochemistry; IF, immunofluorescence.

**Table 2 biology-12-01394-t002:** Primers used in this study.

Primer	Sequence
Glyceraldehyde-3-phosphate dehydrogenase (GADPH)	F: 5′-CAAGGACACTGAGCAAGAGA-3′
	R: 5′-GCCCCTCCTGTTATTATGGG-3′
Tumor necrosis factor (TNF-α)	F: 5′-CCCCCAGTCTGTATCCTTCT-3′
	R: 5′-TTTGAGTCCTTGATGGTGGT-3′

## Data Availability

No new data was created.

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
