# Peer review of "NF-kB Regulation and the Chaperone System Mediate Restorative Effects of the Probiotic Lactobacillus fermentum LF31 in the Small Intestine and Cerebellum of Mice with Ethanol-Induced Damage"

_biology, 2023, doi:10.3390/biology12111394_

Round 1

Reviewer 1 Report (Previous Reviewer 1)

Comments and Suggestions for Authors

Reviewer comments and suggestions

The authors in this study evaluate the mechanisms involved in the putative protective effects of L. fermentum against oxidative stress and inflammation. For this research, mice were fed with ethanol alone or in combination with the probiotic Lactobacillus fermentum (L. fermentum) for 12 weeks. The study reported that the protective effects were due to the modulation of the NF-κB signaling pathway with induction of Hsp60, Hsp90, and IkB-α by the probiotic occurred in the jejunum. L. fermentum inhibited IL-6 expression and downregulated TNF-α transcription. Moreover, this valuable effect of the probiotic on the intestine was repeated on the cerebellum, where downregulation of glial inflammation-related markers was observed in the probiotic-fed mice. The data show that L. fermentum exerted anti-inflammatory and cytoprotective effects in both the small intestine and the cerebellum. 

Overall, the manuscript needs to be modified. I have listed the concerns/comments that needed to be explained/modified. 

  1. Line 71-72 It would be nice if the authors could briefly discuss these cited references (4-7)
  2. Line 85 is there was any report of anti-inflammatory properties of this Lactobacillus fermentum bacteria.
  3. Line 176 a typo error is present, in fig 3 the control and ethanol group the level of Hsp was elevated (seems similar)
  4. Figure 5 E What about the significance between the control and probiotic-treated
  5. Line 565-566 no need to mention these lines at the starting part of the discussion part
  6. Line 568 did the authors discuss any marker of oxidative stress
  7. Line 581-583 Please explain it in a better way
  8. Line 664-665 Please mention the respective figure for a clear understanding
  9. Line 700-701 (46-49) which probiotic these manuscripts have used, kindly explain it

Author Response

Comments and Suggestions for Authors

Reviewer #1 Comment #1 (R#1C#1): Line 71-72 It would be nice if the authors could briefly discuss these cited references (4-7)

Author Reply (AR): Thank you for the suggestion, we modified the text accordingly.

 R#1C#2: Line 85 is there was any report of anti-inflammatory properties of this Lactobacillus fermentum bacteria.

AR: Yes, there are reports of anti-inflammatory properties of Lactobacillus fermentum bacteria of different strain, as reported for example in cited reference 13, but not for the strain LF31. This strain was used in our experiments.

R#1C#3: Line 176 a typo error is present, in fig 3 the control and ethanol group the level of Hsp was elevated (seems similar).

AR: We corrected the typo. Regarding Figure 3, we conducted a semi-quantitative analysis which revealed the Hsps decrease as shown in the histogram.

R#1C#4: Figure 5 E What about the significance between the control and probiotic-treated

AR: There is no significant difference between the control and probiotic-treated mice.

R#1C#5: Line 565-566 no need to mention these lines at the starting part of the discussion part

AR: Thanks for the suggestion. Lines 566-569 are the legend of Figure 11.  

R#1C#6: Line 568 did the authors discuss any marker of oxidative stress

AR: Thank you for this comment. All our work is focused on oxidative stress markers included in NF-kB pathway as Hsp60 and Hsp90.

 R#1C#7: Line 581-583 Please explain it in a better way

AR: Thank you, we modified the text hoping that it is better than before.

R#1C#8: Line 664-665 Please mention the respective figure for a clear understanding

AR: Thank you for the comment, we modified the text accordingly.

R#1C#9: Line 700-701 (46-49) which probiotic these manuscripts have used, kindly explain it

AR: We have added the requested information.

Reviewer 2 Report (Previous Reviewer 2)

Comments and Suggestions for Authors

I still have my concerns related to sample numbers, discussion of the results as I have written in my first review.

Comments on the Quality of English Language

English language can be edited moderately.

Author Response

Comments and Suggestions for Authors

Reviewer #2 Comment #1 (R#1C#1): I still have my concerns related to sample numbers, discussion of the results as I have written in my first review.

Author Reply (AR): We are sorry for this comment but as we have written previously, we chose a size of 5 animals per group in response to a specific request from the ethics committee, which determined the minimum number of animals to be used. This choice followed existing literature and it is in line with the number of animals per group used in a previous publication dealing with the same topic. (see references “Barone, R., Rappa, F., Macaluso, F., Caruso Bavisotto, C., Sangiorgi, C., Di Paola, G., Tomasello, G., Di Felice, V., Marcianò V, Farina F, Zummo G, Conway de Macario E, Macario AJL, Cocchi M, Marino Gammazza, A. Alcoholic liver disease: A mouse model reveals protection by Lactobacillus fermentum. Clin Transl Gastroenterol. 2016, 7(1), e138”). We want underline that the present work is a continuation of the previous one, which included this sample number. We hope to have the opportunity to continue the work investigating other molecules of the pathway and studying other organs increasing the mice number in a new experimental design.  

 R#2C#2: English language can be edited moderately.

AR: Thank you for the comment. The work was carefully reviewed and edited by native English speakers co-authors.

Reviewer 3 Report (Previous Reviewer 3)

Comments and Suggestions for Authors
Dear authors,
The modifications previously indicated were made, as well as a justified response to the comments and questions asked about the work, so I consider that the work can be published in the form in which it is presented.

Author Response

Comments and Suggestions for Authors

Reviewer #3 Comment #1 (R#1C#1): Dear authors, the modifications previously indicated were made, as well as a justified response to the comments and questions asked about the work, so I consider that the work can be published in the form in which it is presented.

Author Reply (AR): We thank very much the reviewer for this comment.

This manuscript is a resubmission of an earlier submission. The following is a list of the peer review reports and author responses from that submission.

Round 1

Reviewer 1 Report

Comments and Suggestions for Authors

Reviewer comments and suggestions

The authors in this study investigated the beneficial effect of a probiotic “ Lactobacillus fermentum” for this they took mice that were fed with ethanol alone or in combination with the probiotic Lactobacillus fermentum (L. fermentum) for 12 weeks. They have studied NF-κB signaling pathway with induction of Hsp60, Hsp90, and IkB-α by the probiotic occurred in the small intestine. The study shows the beneficial effect of the probiotic on the intestine and the cerebellum. The data show that L. fermentum exerted anti-inflammatory and cytoprotective effects in both the small intestine and the cerebellum, by different mechanisms.

Overall, the manuscript was well written. However, a few concerns/comments needed to be explained/modified. 

  1. Line 48-50, a typo error was present, and why two times reference 1 was highlighted
  2. Line 61-62 Please explore these studies (8,10,11)
  3. Line 68-69 Please explain the metabolic disorder related to this
  4. Line 80-84 It would be nice if the authors could present the study design in a ray diagram.
  5. Comments for Figure 1 All figures should be named A, B, C, D, E for clarity and Figure 2 also required an explanation 
  6. Comments for Figure 4 Little confusing, some time they say MMP9 and then TNF alpha.. please check it again
  7. Please modify B with a different one (distorted) and a typo error in line 293
  8. Comments for discussion: The first paragraph should discuss about the novelty of this study rather than discuss other published studies.
  9. Line 377 Please name the Hsps discussed in this study
  10. All references should be modified based on MDPI journal guidelines.

Author Response

Reviewer #1 Comment #1 (R#1C#1): Line 48 50, a typo error was present, and why two times
reference 1 was highlighted
Author Reply (AR):
Thank you for the suggestion, we have corrected the typo and eliminated the
extra reference (see line 51 53)
R#1C#2:
L ine 61 62 Please explore these studies (8,10,11)
AR:
Thank you for the suggestion, but we do not understand why we have to explore the studies. It is
necessary to comment them in extenso ? Or you do not consider them appropriate ly discussed ? Please
let us know.
R#1C#3:
Line 68 69 Please explain the metabolic disorder related to this .
AR:
Thanks for the sugges tion, we have improved the description (see line 71 73)
R#1C#4:
Line 80 84 It would be nice if the authors could present the study design in a ray diagram
AR:
Thanks for the suggestion, we have better explained the animal model in the methods section and
created an explanatory image (see Figure 1)
R#1C#5:
Comments for Figure 1 All figures should be named A, B, C, D, E for clarity and Figure 2
also required an explanation
AR:
Thanks for the suggestion please see the new version revised following it .
R#1C#6:
Comments for Figure 4 Little confusing, some time they say MMP9 and then TNF alpha.
P lease check it again
AR:
We checked it accordingly.
R#1C#7:
Please modify B with a different one (distorted) and a typo error in line 293
AR:
Thank you, we have c orrected text
R#1C#8:
Comments for discussion: The first paragraph should discuss about the novelty of this study
rather than discuss other published studies.
AR:
Thank you for the comment, we modified the text accordingly.
R#1C#9:
Line 377 Please name the Hsps discussed in this study
AR: We have added the requested information (see line
R#1C#10:
All references should be modified based on MDPI journal guidelines
AR:
We apologize for the mistake we have fixed all references according to the journal ’s rules.

Reviewer 2 Report

Comments and Suggestions for Authors

The topic of the paper is interesting, the chapters of the manuscript are well-structured, the paper is generally clear. However, I have some concerns and explain them in more detail below. I ask that the authors specifically address each of my comments in their responses (see the attached file).

Comments on the Quality of English Language

Moderate editing of English language can be done. 

Author Response

Reviewer #2 Comment #1 (R#1C#1):
One of my main concerns is that the study was carried out with
5 mice per treatment. In my opinion, the number of animals is very small, and molecular genetic
methods are usually carried out with 8 10 animals, but with 6 at least. For some measurements (e.g.
villus traits) this number would be fair enough, but not for gene or protein expression measurements.
If the paper allows to p ublish brief communication, I would suggest publishing in that form.
Author Reply (AR):
We thank the reviewer for this co mment . T he number of 5 animals per group was chosen as a result of a
specific request of the ethic s committee concerning the minimum number of
animals to be used We based that on literature data and it is the same number of animals per group we
have previously used in a published work on the same topic (see references Barone, R., Rappa, F.,
Macaluso, F., Caruso Bavisotto, C., Sangiorgi, C., Di Paola, G., Tomasello, G., Di Felice, V., Marcianò V, Farina F, Zummo G, Conway de Macario E, Macario AJL, Cocchi M, Marino Gammazza, A.
Alcoholic liver disease: A mouse model reveals protection by Lactobacillus fermentum. Clin Transl
Gastroenterol. 2016, 7(1), e138”
R#2C#2:
My other concern is that the authors examine the beneficial effect of L. fermentum on the
intestinal damage induced by ethanol treatment. In my opinion, the probiotic supplemented EtOH
group should be compared to the EtOH group statistically instead of the basic control group. In this
case, control treatment serves to observe the harmful effect of EtOH treatment. I suggest comparing
and discussing the results as I mentioned.
AR:
Thank you ver y much for the suggestion. These experiments are part of big project from which
we already published a paper containing the comparison between the same groups please see the
references included in the manuscript “Barone, R., Rappa, F., Macaluso, F., Carus o Bavisotto, C.,
Sangiorgi, C., Di Paola, G., Tomasello, G., Di Felice, V., Marcianò V, Farina F, Zummo G, Conway de Macario E, Macario AJL, Cocchi M, Marino Gammazza , A. Alcoholic liver disease: A mouse model
reveals protection by Lactobacillus fermentum. Clin Transl G astroenterol. 2016, 7(1), e138 Since we
consider the present work as a continuation of the previous, we would like to maintain the same sc heme
of anal ysis.
R#2C#3:
The reader cann ot conclude which small intestinal part was used for measurement. Clarify
which small intestinal part and which segment was applied for measurements, please.
AR:
We used je ju nu m and we modified the text accordingly.
R#2C#4:
Authors have cut and stained bea utiful intestinal segments but only measured the villus
height. There are many histomorphological parameters (e.g. villus width, crypt depth, VH: CD ratio,
total mucosa thickness, etc.) that could be measured in these photos. In other cases, I also feel th at the
authors did not go deep enough in measurements. E.g., they purified RNA from tissues but only
measured TNF gene expression. Other pro inflammatory cytokines could be also measured at the
mRNA level.
AR:
We thank the reviewer for these suggestions Regarding the histomorphological parameters , we
added other measurements as supplementary information while for o ther pro inflammatory cytokines
from RNA, w e are aware that there are many molecules to be investigated. We would like to continue
this project , then we hope to go deep er in a further work.
R#2C#5:
In my opinion, the Discussion chapter sometimes does not conclude the results properly.
Authors discu ss, that (lines 315 317) The results obtained show that the probiotic can preserve the
intestinal barrier structure and villus morphology in mice receiving ethanol and the probiotic when
compared with mice receiving only ethanol. From which results can the authors conclude this? As I
see, villus height was higher in Et OH (damaged group), while it was lower in the probiotic treated
group compared to the Et OH group. As higher villi the beneficial is, I cannot understand the authors
conclusion. Results and d iscussion are also controversial in HSP60 and HSP90 distribution. They are
usually higher during intestinal damage, while their expression decrease in case of improvement.
AR:
Thank you for this comment. The conclusion that “ The probiotic can preserve the intestinal barrier
structure ” belong to the results regarding the increase of ZO1 (a protein of the tight junctions observed by immunofluorescence and the decrease of MMP9 and
NF kB (involved in the inflammatory response
having as a target th e barrier) observed by immunohistochemistry after probiotic intake. Higher villi is
not beneficial, and the probiotic seems to have the capacity to reduce villi height preserving then their
morphology . Regarding Hsp60 and H sp 90 they are a critical component of the stress response for the
maintenance of tissue homeostasis . In our opinion, their expression increased after probiotic intake to
counteract the effect of et hanol on the intestinal barrier. H sp s have a cytoprotective role in epithelial
cell function and structure see reference 8, 10 and 11) The induction of H sp by intestinal flora (or as
in this case by the probiotic) serves to maintain the barrier function and to protect the host from the
injurious effects of agents such as reactive oxygen species and ethanol
R#2C#6:
In Material and Methods, histopathological part: what was measured? It is not described.
How many vi lli were measured?
AR:
We modified the text. Regarding the number of villi, we measured 10 villi for each mouse, then
50 villi for each group.
R#2C#7:
RNA preparation and qRT PCR Assay: Which RNA isolation kit was applied? Was DNase
treatment applied? How were RNA quality and quantity determined? By which spectrophotometer?
Provide manufacturers, please
AR:
We completed the text according to the reviewer's suggestion (see line 172 179)
R#2C#8:
Line 87: what type of ethanol was applied? Purity, manufacturer?
AR:
We have added the requested information, as can be seen in line 96
R#2C#9:
In my opinion, the probiotic treated group should be named, such as EtOH+ L.fermentum or
probiotic. In this form “P” can be hardly recognized. You can also leave “12” from the name of the
treatments.
AR:
We ch anged the text according to the suggestion. Nonetheless, for aesthetic purposes and to
enhance the clarity of the image, we choose to include 12EtOH P in the diagrams.
R#2C#10:
TNF is not written on the graph.
AR:
We added TNF α , as can be seen in the graph in figure 5E.

Reviewer 3 Report

Comments and Suggestions for Authors

REVIEW

Dear authors,

Please consider the following comments to improve the content of your manuscript before publication. 

MAJOR CONCERN:

-        In section 2.1 Animal model, more information is missing about the experimental scheme used for the inoculation of ethanol (EtOH), they mention that it was administered for 12 weeks, however, they do not indicate the volume inoculated per mouse, what was the percentage of EtOH administered per mouse? Was the EtOH administered daily for 12 weeks or how many times per week? They should describe this experimental scheme in detail.

-        They omit the details of the treatment with L. fermentum LF31. Was the treatment with L. fermentum LF31 daily or how many times a week? How do you ensure that a mouse ingested the amount of CFU indicated (109) in the water? The best way to ensure the amount of CFU is intragastrically.

-        It was not necessary to add the control group that received the treatment only with L. fermentum LF31, due to the time of administration of the probiotic (12 weeks) it must be monitored that it is innocuous in the mice.

-        They did not monitor the weight of the mice during the development of the experiment.

-        The number of animals used is very small (n=5), do you plan to increase the number?

-        Did they mention that they used 12-month-old mice, are they evaluating senescence or are they 12-week-old mice?

-        Why didn't they carry out evaluations in the liver? It is one of the organs that are mainly affected by alcohol intake, at least histopathology. Another organ commonly used for histopathological analysis is the spleen, which could complement the results presented.

-        In the discussion, they mention that there are intestinal bacterial components such as LPS or PGN that can reach distant organs through systemic circulation. These 2 biomarkers of endotoxemia in serum could be evaluated to demonstrate the protection of treatment with L. fermentum LF31 at the level of intestinal permeability .

MINOR CONCERN:

-        In the title of the work they must indicate the name of the strain of L. fermentum (LF31).

-        In the Introduction they do not mention the strain used in the work (L. fermentum LF31), if it is a commercial probiotic they must describe the origin of the strain and some characterized function.

-        The size of the histograms in Figures 1B, 2, 4A, 4B and 5B are very small, enlarge the images.

-        Check that the text of the significant differences in Figures 4B, 4C and 5B correspond to the images, it seems that the * do not correspond.

-        In Figure 1 they mention that there was a reduction in Goblet cells, how they carried out the analysis, since they only present the histogram of the length of the villi.

-        Homogenize the metalloproteinase 9 script (MMP-9).

It is necessary to make the following corrections in the indicated lines:

Line 70: write in cursive “Lactobacillus”.

Line 83: add a space in “Health(authorization”.

Line 163: add a space in “380mm”, “600mm” y “500mm”.

Line 166: add a space in “Figure1B”.

Lines 173, 175: add a space in “100mm” y “20mm”.

Line 189: add a space in “400X,scale”.

Line 197: add a space in “andmetalloproteinase”.

Line 198: add a space in “immunofluorescence,while”.

Line 217: add a space in “Figure4”.

Line 232: add a space in “onpNF-kB”.

Line 235: add a space in “andIkB-a”.

Line 242: capitalize “pNF-kb”.

Line 257: add a space in “observationsand”.

Lines 264, 274: add a space in “12EtOH” y “12EtOH-P”.

Line 293: add a space in “ofHsp60”.

Line 298: add a space in “EtOH,in” y “WM(Figure”.

Line 342: add a space in “alcohol(e.g.,”.

Line 353: add a space in “EtOH-Pgroup”.

Line 354: add a space in “12EtOH”.

Line 359: add a space in “andS100b”.

Line 373: remove a space in “speculate  that”.

Line 381: add a space in “inhibitor)in”.

Line 417: add a space in “restoredthroughMMP9”.

References

Review the format of the magazine and correct, as well as write the name of the microorganisms in italics.

Please amend the requested comments and submit the revision file.

Author Response

Reviewer #3 Comment #1 (R#1C#1):
In section 2.1 Animal model, more information is missing about
the experimental scheme used for the inoculation of ethanol (EtOH), they mention that it was
administered for 12 weeks, however, they do not indicate the volume inoculated per mouse, what was
the percentage of EtOH administered per mouse? Was the EtOH administ ered daily for 12 weeks or
how many times per week? They should describe this experimental scheme in detail
Author Reply (AR):
We thank the reviewer for this comment. W e have added more detail s in the text.
R#3C#2:
They omit the details of the treatment with L. fermentum LF31. Was the treatment with L.
fermentum LF31 daily or how many times a week? How do you ensure that a mouse ingested the
amount of CFU indicated (10 9 ) in the water? The best way to ensure the amount of CFU is
intragastrically.
AR:
We thank the reviewer for this comment. W e have added more details in the methods section of
the manuscript . We are aware that the intragastric administration could be the best way, but we do not
have the equipment to do it. Then, we decided to fed the mic e manually with a pipette and the y were
particularly predisposed in ingesting the probiotic as probably they found it pleasant tasting .

R#3C#3:
It was not necessary to add the control group that received the treatment only with L.
fermentum LF31, due to the time of administration of the probiotic (12 weeks) it must be monitored
that it is innocuous in the mice.
AR:
These experiments are part of big project from which we already published a paper (see reference
included in the manuscript “ Barone, R., Rappa, F., Macaluso, F., Caruso Bavisotto, C., Sangiorgi, C.,
Di Paola, G., Tomasello, G., Di Felice, V., Marcianò V, Farina F, Zummo G, Conway de Macario E, Macario AJL, Cocchi M, Marino Gammazza, A. Alcoholic liver disease: A mouse model reveals
protection by Lactobacillus fermentum. Clin Transl G astroenterol. 2016, 7(1), e138” in which we did
not included the treatment only with L. fermentum LF31 . We thank th e reviewer for this suggestion,
and we hope to have to opportunity to include this group in a new experimental design.
R#3C#4:
They did not monitor the weight of the mice during the development of the experiment.
AR:
Thank you for this comment. Food and water intake were measured daily and body weight was
measured three times a week. The obtained data showed no significant diffe rences within and between
groups . As mentioned above, t hese experiments are part of big project from which we already published
a paper and food, water intake as well as body weight measurements were reported in the
"supplementary data" section of the references included in the manuscript “ Barone, R., Rappa, F.,
Macaluso, F., Caruso Bavisotto, C., Sangiorgi, C., Di Paola, G., Tomasello, G., Di Felice, V., Marcianò V, Farina F, Zummo G, Conway de Macario E, Macario AJL, Cocchi M, Marino Gammazza, A.
Alcoholic liver disease: A mouse model reveals protection by Lactobacillus fermentum. Clin Transl
Gastroenterol. 2016, 7(1), e138 ”.
R#3C#5:
The number of animals used is very small (n=5), do you plan to increase the number?
AR:
We thank the reviewer for the consideration. T hese experiments are part of big project from which
we already published a paper (see R#3C#4 ) and the number of 5 animals per group was the result of a
specific request of the ethics committee concerning the minimum n umber of animals to be used.
R#3C#6:
Did they mention that they used 12 month old mice, are they evaluating senescence or are
they 12 week old mice?
AR:
Thank you for this comment. We are evaluating the effect of the probiotic in the adult population
using ethanol as oxidative stress and inflammatory inducer with the purpose to use these results for the
evaluation of the probiotic efficacy in different model of bowel inflamm atory disease Considering the
table reported by the Jackson Laboratory https://www.jax.org/news and insights/jax
blog/2017/november/when are mice cons idered old )), m ice of 1 0 1 4 months correlates to humans from
38 47 years old. Since it has been reported that most of the diagnosis for inflammatory bowel diseases
are in the adult population, we think that 12 month old mice are the right choice.
R#3C#7:
Why didn't they carry out evaluations in the liver? It is one of the organs that are mainly
affected by alcohol intake, at least histopathology. Another organ co mmonly used for histopathological
analysis is the spleen, which could complement the results presented.
AR: We thank the reviewer very much for this comment. Actually, we have already published the data
obtained on t he liver. Please see “Barone, R., Rappa, F., Macaluso, F., Caruso Bavisotto, C., Sangiorgi, C., Di Paola, G., Tomasello, G., Di Felice, V., Marcianò V, Farina F, Zummo G, Conway de Macario E, Macario AJL, Cocchi M, Marino Gammazza, A. Alcoholic liver disease: A mouse model reveals protection by Lactobacillus fermentum. Clin Transl Gastroenterol. 2016, 7(1), e138 ”. In this new paper we directed our attention to the communication between the gut and the brain after the induction of oxidative stress and inflammation in the gut.

R#3C#8:
In the discussion, they mention that there are intestinal bacterial components such as LPS or
PGN that can reach distant organs through systemic circulation. These 2 biomarkers of endotoxemia
in serum could be evaluated to demonstrate the protectio n of treatment with L. fermentum LF31 at the
level of intestinal permeability.
AR:
We thank the reviewer for this important suggestion . We hope to have the opportunity to do this
evaluation in the future .
R#3C#9:
In the title of the work they must indicate the name of the strain of L. fermentum (LF31).
AR:
We have modified the title according to this request.
R#3C#10:
In the Introduction they do not mention the strain used in the work (L. fermentum LF31),
if it is a commercial probiotic they must describe the origin of the strain and some characterized
function.
AR:
We modified the text according to the reviewer’s request (see line
R#3C#11:
The size of the histograms in Figures 1B, 2, 4A, 4B and 5B are very small, enlarge the
images.
AR:
Thanking the reviewer for this suggestion, we have made the requested changes.
R#3C#12:
Check that the text of the significant differences in Figures 4B, 4C and 5B correspond to
the images, it seems that the * do not correspond.
AR:
Thanking the reviewer for this suggestion, w e modified the text according to the reviewer’s
requests (see figure 5D and
R#3C#13:
In Figure 1 they mention that there was a reduction in Goblet cells, how they carried out
the analysis, since they only p resent the histogram of the length of the villi.
AR:
The evaluation of goblet cells was conducted semi quantitatively through a morphological
observation as reported in the text (“A reduction in the number of globet cells… was also observed…”)
by two pathologists independently and in a double blind manner.
R#3C#14:
Homogenize the metalloproteinase 9 script (MMP 9).
AR:
Thank you, we have modified the tex t according to the request.
R#3C#15:
It is necessary to make the following corrections in the indicated lines:
Line 70: write in cursive “Lactobacillus”.
Line 83: add a space in “Health(authorization”.
Line 163: add a space in “380mm”, “600mm” y “500mm”.
Line 166: add a space in “Figure1B”.
Lines 173, 175: add a space in “100mm” y “20mm”.
Line 189: add a space in “400X,scale”.
Line 197: add a space in “andmetalloproteinase”.
Line 198: add a space in “immunofluorescence,while”.
Line 217: add a space in

Line 232: add a space in “onpNF
kB”.
Line 235: add a space in “andIkB
a”.
Line 242: capitalize “pNF
kb”.

Line 257: add a space in “observationsand”.
Lines 264, 274: add a space in “12EtOH” y “12EtOH
P”.
Line 293: add a space in

Line 298: add a space in “EtOH,in” y “WM(Figure”.
Line 342: add a space in “alcohol(e.g.,”.
Line 353: add a space in “EtOH
Pgroup”.
Line 354: add a space in “12EtOH”.
Line 359: add a space in “andS100b”.
Line 373: remove a space in “speculate t
hat”.
Line 381: add a space in “inhibitor)in”.
Line 417: add a space in “restoredthroughMMP9”.
Review the format of the magazine and correct, as well as write the name of the microorganisms in
italics.
AR:
With apologies for the typos, we have corrected th e m as indicated .